# The Balance in the Head: How Developmental Factors Explain Relationships Between Brain Asymmetries and Mental Diseases

**DOI:** 10.3390/brainsci15020169

**Published:** 2025-02-09

**Authors:** Martina Manns, Georg Juckel, Nadja Freund

**Affiliations:** 1Research Division Experimental and Molecular Psychiatry, Department of Psychiatry, Psychotherapy and Preventive Medicine, LWL University Hospital, Ruhr-University, 44809 Bochum, Germany; nadja.freund@rub.de; 2Department of Psychiatry, Psychotherapy and Preventive Medicine, LWL University Hospital, Ruhr-University, 44791 Bochum, Germany; g.juckel@lwl.org

**Keywords:** cognitive impairment, gene environment, lateralisation, ontogeny, stress, translational/preclinical research

## Abstract

Cerebral lateralisation is a core organising principle of the brain that is characterised by a complex pattern of hemispheric specialisations and interhemispheric interactions. In various mental disorders, functional and/or structural hemispheric asymmetries are changed compared to healthy controls, and these alterations may contribute to the primary symptoms and cognitive impairments of a specific disorder. Since multiple genetic and epigenetic factors influence both the pathogenesis of mental illness and the development of brain asymmetries, it is likely that the neural developmental pathways overlap or are even causally intertwined, although the timing, magnitude, and direction of interactions may vary depending on the specific disorder. However, the underlying developmental steps and neuronal mechanisms are still unclear. In this review article, we briefly summarise what we know about structural, functional, and developmental relationships and outline hypothetical connections, which could be investigated in appropriate animal models. Altered cerebral asymmetries may causally contribute to the development of the structural and/or functional features of a disorder, as neural mechanisms that trigger neuropathogenesis are embedded in the asymmetrical organisation of the developing brain. Therefore, the occurrence and severity of impairments in neural processing and cognition probably cannot be understood independently of the development of the lateralised organisation of intra- and interhemispheric neuronal networks. Conversely, impaired cellular processes can also hinder favourable asymmetry development and lead to cognitive deficits in particular.

## 1. Brain Lateralisation

Cerebral lateralisation, which refers to the differential role of the left and right hemispheres for specific functions, is a fundamental organisational principle of our brain that has a decisive influence on neuronal processing and cognitive performances. Thinking about the consequences of having two differentially acting brain hemispheres has a long tradition rooted in classical neuroanatomy. It was the groundbreaking claim of Paul Broca during a meeting of the French Anthropological Society, “Nous parlons avec l’hemisphere gauche”, that demarcated the dawn of asymmetry research in 1861 [1,2]. Since that time, research has shown that not only language processing but most aspects of information processing, cognition, and behavioural control include asymmetrical components. This means that the two hemispheres differ structurally and functionally and are, therefore, involved to different degrees in certain mental processes. For example, a majority of people show a left-hemispheric dominance for language processing and fine motor control (and, thus, for right-handedness), while facial recognition and aspects of emotional and spatial processing are dominated by the right hemisphere [2,3,4]. It has long been debated whether cerebral lateralisation is a specific human trait, but studies from the field and laboratory have shown that left–right asymmetries in the brain and of behaviour are widespread all across the animal kingdom [2,5,6]. This common distribution suggests the profound benefit of a lateralised brain, presumably because such an organisation allows for parallel and complementary processing in each hemisphere that, in turn, saves neuronal resources and optimises the processing speed [5,6].

Functional asymmetries are related to structural left–right differences between the two halves of the brain at the cortical and subcortical levels [2,3,7]. In humans, there areneuroanatomical asymmetries in both white and grey matter. In addition to the size and shape asymmetries of cortical gyri and sulci [8,9,10,11], there are hemispheric differences in connectivity [12,13], as well as cellular and molecular organisation [14,15,16]. These structural left–right differences are probably optimised for specific processing modes that are relevant for the conductance of certain cognitive tasks. For instance, the left-hemispheric dominance in language processing is probably caused by an advantage in encoding rapid frequency transitions [17]. Consequently, lesions of the left and right hemispheres result in different functional impairments, so that cerebral asymmetries affect the symptoms and recovery of stroke patients [18] just as the classic cases of Broca firstly indicated [1]. However, recent research indicates Broca’s aphasia is not solely based on damage to the classical Broca area but also requires damage to the adjacent grey and white matter due to the complex cortical and subcortical connectivity pattern [19,20]. In neurodegenerative diseases, the left and right hemispheres can be affected differently as the disease progresses [21,22]. For instance, the typical thinning of higher-order cortical regions, which results in a progressive loss of asymmetry over the lifespan, is accelerated in Alzheimer’s patients [23]. In Parkinson’s disease, lateralised motor symptoms are the result of an asymmetrical pattern of neurodegeneration in the brain [24,25], whereby the asymmetrical DNA methylation patterns of cortical neurons are associated with progressive symptoms [26].

## 2. Brain Lateralisation and Pathophysiology of Mental Disorders

It is noteworthy that there appears to be a prototypical division of labour between the hemispheres in the human brain, such that the left and right hemispheres are dominant for the same functions (e.g., left-hemispheric dominance for language processing and right-hemispheric dominance for visuospatial processing) in the majority of the population, though there is a wide interindividual variation in hemispheric specialisation for certain functional domains [4,27,28,29], and within one functional domain, sub-processes can be distributed to different hemispheres (e.g., language networks typically show left-hemispheric dominance for language production and decoding and right-hemispheric dominance for prosody analysis [2,30]). Despite these variations, the human-specific alignment of hemispheric functional dominances has led to the idea that the population-typical pattern is beneficial for optimal cognitive processing and superior mental abilities [4,31,32,33], while a disorganised, lateralised organisation could lead to cognitive deficits [34]. In line with this idea, persons with intellectual disabilities show an elevated level of atypical handedness [35]. The typical pattern of structural and functional asymmetries is also altered in neurodevelopmental disorders and psychiatric illnesses. The asymmetries are usually reduced or no longer present but can sometimes be more pronounced [10,36,37,38,39,40]. It is striking that cognitive deficits in particular, e.g., in the areas of learning and memory or executive functions [41,42], which have lateralised processing components in the healthy population [43], show an altered lateralisation pattern [37,38,39], with unusual patterns being related to the severity of cognitive impairment or the duration of the disease [44,45]. The diverse mental disorders display different patterns and degrees of altered brain asymmetries for certain functional domains, which can be detected at the behavioural or functional network activity level.

The most explored area of study is preferential hand use (i.e., handedness) as an apparent indicator for primarily lateralised action control but also for functional brain lateralisation in general [46]. Handedness is related to other behavioural asymmetries, like preferential foot use [47] or a cradling bias [48]. Since hand use is controlled by the contralateral brain hemisphere, a right or left hand preference implies left- or right-hemispheric dominance, respectively [49]. In humans, about 90% of individuals prefer to use the right hand for complex manual tasks, like fine-tuned object manipulation or writing, while a minority displays left-handedness (=preferential left hand use), mixed-handedness (=differential hand use for different motor acts), or non-right-handedness (=no hand dominance) [49]. Left-handedness, per se, is not a negative trait, but non-right-handedness or mixed-handedness possibly reflect the disturbed neurodevelopment of cortical brain areas. Since handedness manifests early during ontogeny, a weakened dominance for action control may indicate the susceptibility to develop a pathophysiology before the onset of a disorder [35]. Patients with schizophrenia, for example, display a higher prevalence of non-right-handedness [50,51] and generally show poor neuromotor function even before a first psychotic episode [52]. Non-right-handedness is also increased in autism spectrum disorder (ASD) [53,54], attention deficit hyperactivity disorder (ADHD) [55], post-traumatic stress disorder [56], and clinical subtypes of bipolar disorder [51,57]. No association could be identified for depression [46,58] or anxiety disorders, at least in adult patients [59].

In schizophrenia, reduced language lateralisation can be observed, especially in patients experiencing auditory hallucinations [60,61]. Moreover, patients show the untypically lateralised processing of working memory and attention [62], the impairments of which are core deficits in schizophrenia [52,63]. In ASD, the lateralisation of language processing is also modified [64]. In addition, abnormal functional asymmetries have been reported in cortical networks, which process motor, somatosensory, visuospatial attention, and working memory [64,65,66]. Individuals with ASD may exhibit highly individualised patterns of both extreme rightward and leftward deviations from the typical patterns that are associated with symptom severity [67]. In ADHD, there is a profound changes in resting-state functional magnetic resonance imaging in different cortical areas, which are related to inattentive and hyper/impulsive scores [68,69]. In depression, studies based on acoustic paradigms indicate the modified lateralisation of perceptual and emotional processing [70]. Regarding bipolar disorder, hints at changes in language lateralisation or other behavioural asymmetries are mixed in the literature [57]. However, neurophysiological and functional imaging studies indicate that mania is related to increased left-hemispheric neuronal activity and depression to the enhanced right-hemispheric activity of specific brain regions in the frontal and temporal lobe [71].

Apart from functional specialisations, there are several left–right differences in the organisation of cortical and subcortical grey and white matter, which are modified in mental disorders [9,72,73] and which may reflect changes at the molecular, cytoarchitectonic, or network level [39,74].

In schizophrenia, there are changes in the left–right differences of cortical thickness, surface area, and subcortical volumes in the brain regions associated with emotion, memory, and visual processing [75], and particularly in language-related cortical regions [74]. In addition, intra- and interhemispheric white matter connections are modified, with significant decreases in both the global and nodal efficiency of hemispheric networks compared to healthy controls, mainly in frontal and temporal regions [74]. In ASD, asymmetries of grey matter are reduced in regions that are widely distributed across the cortex [75,76]. Thereby, right-hemispheric neuronal networks subserving executive functions, as well as language-related and sensorimotor processes, display altered connectivity and network efficiency. Atypical asymmetry patterns can be associated with communication symptoms and non-verbal intelligence [44,77,78]. Modified lateralisation patterns are not necessarily constant over a lifetime. For instance, paediatric, but not adult, obsessive-compulsive disorder is associated with altered subcortical asymmetry [79]. In ADHD, only small age-specific changes of brain asymmetries can be observed. Apart from some differences in cortical thickness asymmetry across different age groups, there is less rightward asymmetry of the total hemispheric surface area specifically in children. In adults with ADHD, globus pallidus asymmetry is altered compared to those without ADHD [80]. In addition, there are hints at reduced hemispheric asymmetries in white matter properties, which are related to the clinical features of ADHD [81]. There are some reports for modified volume asymmetries in limbic structures in patients with post-traumatic stress disorder [82] and major depression disorder [83,84]. However, in depression, modified asymmetries in cortical and subcortical areas cannot be consistently identified, despite the presence of neuronal activity differences between the hemispheres [85,86]. It is possible that aversive experiences, like stressful life events, critically affect volumetric changes [87].

In sum, the degree and extent of changes in the functional and neuroanatomical asymmetry pattern vary between disorders and can affect different cognitive modules. However, it is far from being clear how structural changes are related to modified neuronal processing, functional network properties, and, ultimately, cognition and behaviour. First of all, macroscopical asymmetries in the volume and size of brain areas are not necessarily related to functional lateralisations. There is, for instance, a dissociation between the macroscopic asymmetry of language processing areas, like the planum temporale, and language laterality [12], and there are no clear associations between cerebral cortical asymmetry measures and handedness [9,10]. In actuality, the pattern of left–right differences in cortical thickness and surface areas differ between left- and right-handed people and are not simply mirror images [88]. Moreover, behavioural lateralisation and underlying functional connectivity asymmetries can emerge in the absence of macroscopic left–right differences [89]. Ambiguities are probably due to the fact that macroscopic features such as the surface size, volume, or thickness of brain areas do not provide information about their microscopic organisation. Currently, there is increasing evidence that functional lateralisations result from microanatomical asymmetries, like myelin content, or neurite density and orientation [90,91,92], as well as intra- and interhemispheric white matter connectivity [12,93] and physiological network properties, which can vary depending on situational context or endogenous states [94]. Since we do not know enough about how structural asymmetries at different neuroanatomical levels are related to the functional organisation of brain circuits, we currently do not understand if and how altered lateralisation patterns can be related to the emergence of functional symptoms and cognitive impairments observed in mental disorders.

The ubiquity of altered lateralisation patterns in a wide range of disorders, however, suggests that the observed associations are not random but have a developmental and/or functional basis. The structural and functional organisation of brain networks emerge by aversive or protective gene–environment interactions, so it makes sense to assume that links between brain lateralisation and the pathophysiology of mental disorders are rooted in developmental processes. Potential associations are obvious for neurodevelopmental disorders, such as ASD or ADHD, and indeed these disorders show particularly pronounced changes in the cerebral lateralisation pattern [67,69,76]. Serious psychiatric disorders typically begin in adolescence or young adulthood [95,96]. However, there are often non-specific symptoms, cognitive impairments, special personality traits, or abnormalities in brain organisation, which precede the clinical manifestation of a disorder and, therefore, indicate maladapted brain development and susceptibility towards aversive experiences [63,97,98]. The associations between altered brain lateralisations and the pathogenesis of mental disorders may vary depending on the developmental trajectory and underlying biological causes of a specific disorder [99], but they could potentially act at two levels. On the one hand, ontogenetic differentiation processes may result in impaired structural and/or functional circuits, which are directly relevant to the pathogenesis of a disorder, or they may increase susceptibility to aversive influences that induce the development of a disorder only later in life.

The development of both, brain lateralisations and mental disorders, is determined by complex gene–environment interactions, with aversive biological factors promoting the pathogenesis of a disorder on the one hand and affecting the degree and direction of cerebral asymmetries on the other [39,100] (Table 1). Accordingly, changes in the typical lateralisation pattern may indicate the influence of aversive factors that disrupt the development of brain networks and lead to the clinical features of a pathophysiology. However, it is also conceivable that the asymmetry formation and the pathogenesis of a mental disorder influence each other via intertwined developmental pathways. Recent large-scale studies have indeed found correlations between lateralisation patterns, susceptibility to mental disorders, and genetic factors [44,101,102].

Apart from the primary symptoms, many mental disorders are accompanied by impairments of higher order cognitive functions, like learning and memory, or executive functions, which overlap in different disorders [41]. This overlap suggests that cognitive deficits may not be directly attributable to primary cellular changes in neurotransmitter systems, neuronal signalling, or epigenetic modifications, which account for the neurobiological basis of a specific mental disorder and which can affect all levels of neural function including sensory, emotional, or motivational processing. Higher-order cognitive deficits could only arise as a secondary consequence of impaired neural processing and, thus, may not be easily influenced by treating the primary biological causes of a disease. For instance, in schizophrenia, cognitive impairments respond only poorly to the known medical treatments, which act mainly on catecholaminergic transmitter systems [52,63,192].

Since the direction and extent of hemispheric lateralisation critically contributes to optimal cognitive processing, it is conceivable that atypical lateralisation is specifically involved in the development of cognitive deficits. Thus, if we want to understand the potential associations between asymmetry formation and the pathogenesis of a mental disorder, we must distinguish between a primary neuronal level, on which the biological basis of a disorder manifests, and a secondary level, which results from disturbed neuronal information processing. But what are, generally, the possible ontogenetic mechanisms that link the development of a mental disorder and an altered lateralisation pattern of the brain?

## 3. Ontogenetic Pathways Linking Brain Lateralisation and Pathogenesis of Mental Disorders

It is assumed that many psychopathological disorders are primarily caused by unbalanced transmitter systems and/or disturbed synaptic mechanisms, which have a multifactorial genetic foundation (ADHD [193]; ASD [194]; depression [195,196]; schizophrenia [197,198]; bipolar disorder [196,199,200]). These imbalances lead to disorganised intra- and interhemispheric neuronal networks, which are ultimately responsible for the symptoms of a disorder. The clinical manifestation of a disorder, however, depends on the action of aversive epigenetic influences during different pre- and postnatal developmental phases, which affect neuronal differentiation processes, and impairs the functional brain organisation. Such a sequence of ontogenetic steps is particularly evident in the pathogenesis of schizophrenia, which requires several biologically significant events (or “hits”) distributed throughout prenatal and postnatal development. Prenatal “first hits”, such as genetic predispositions or the infections of pregnant mothers, lead to aberrations in brain development, which result in the onset of the full clinical syndrome during adolescence in reaction to additional aversive experiences [201,202].

In principle, the action of genetic and non-genetic factors, which contribute to a specific pathophysiology, should affect the development of the left and right brain sides in the same way. From the earliest stages of development, however, there are asymmetries between the left and right hemispheres of the brain, with cortical [203,204], subcortical [205,206], and behavioural [207,208] left–right differences emerging already during prenatal development. Structural and functional lateralisations further manifest during postnatal ontogeny and may change over a lifetime due to ongoing developmental processes under genetic control [88,101,209] or the impact of endogenous factors, like fluctuating hormone levels [39,94,210]. This means that the factors responsible for a pathophysiology always affect an asymmetrical nervous system on the one hand and can themselves also influence asymmetrical developmental processes on the other.

The development of the nervous system can be roughly divided into three phases, during which asymmetries in the intra- and interhemispheric communication of neuronal circuits emerge, and the degree of hemispheric specialisation increases [211]. The first phase (embryonic patterning) comprises the earliest embryological steps in which the three axes of the body plan (antero-posterior, dorso-ventral, left–right) are established. The second (neuronal differentiation) phase includes the differentiation of neuronal elements and networks, while the third (network refinement) phase is characterized by the activity-dependent fine-tuning of neuronal circuitries within the scope of synaptic plasticity, which determines the lateralised functional organisation of intra- and interhemispheric processing systems (Figure 1). During the first phase, complex cascades of genetic and epigenetic interactions break the symmetry of the embryo and lead to the lateralised placement of internal organs and the asymmetries of paired organs, such as the nervous system [177,212,213]. As a result, left–right differences in the developmental speed and the number of neuronal elements emerge, whereby in human embryos, the right hemisphere tends to develop a little earlier than the left one [203]. Therefore, even in the earliest stages of brain development, genetic influences can have differential effects on left- and right-sided neuronal cell populations and can lead, among other things, to asymmetrical susceptibility to aversive factors that can trigger the pathogenesis of a mental disorder (Figure 1). As a consequence, impairments of differentiation processes within certain neuronal cell populations or transmitter systems in the second developmental phase have different effects on the left and right sides of the brain and, thereby, may increase lateralised aberrant developmental tendencies. It is well known that, for instance, the dysregulation of glutamatergic, dopaminergic [63,214,215,216,217,218,219], and GABAergic [218,220,221,222,223,224] systems play a role in the pathophysiology of different disorders. Consequently, dysfunctional neuronal networks might develop, the impairments of which may stabilize over the course of synaptic refinement processes. During this phase, cellular mechanisms mediating activity-dependent plasticity are critical. There is some evidence that these mechanisms are compromised in mental disorders. For instance, ASD is associated with mutations in the SCN2A gene, which lead to the reduced expression of the voltage-dependent sodium channel Na_V_1.2 and, therefore, attenuate the backpropagation of action potentials into the dendrites of cortical neurons, preventing spike-timing-dependent synaptic plasticity [225]. Another example is N-methyl-D-aspartate receptors (NMDAR), which are glutamate-gated ion channels playing critical roles in brain development and plasticity [226], and NMDAR dysfunction plays a role in different mental disorders [227,228]. For example, schizophrenia is characterized by NMDA hypoactivity in cortical (especially in parvalbumin-positive) GABAergic interneurons [222,228,229].

In sum, the described developmental steps can ultimately lead to differential contributions of left- or right-hemispheric neuronal circuitries to the pathophysiology of a mental disorder and/or accompanying cognitive impairments but also therapeutical effects. Conversely, factors that primarily increase the risk of developing a mental disorder may also affect the developmental steps through which asymmetric neuronal structures or functions manifest. The exact cellular mechanisms and the extent and direction of possible interactions may vary depending on the mental disorder, affected cell populations, or functional subsystems and developmental stages. In principle, however, there are at least four conceivable ways in which the developmental pathways of brain lateralisation and mental disorders as two separated traits can be linked [211] (Figure 2).

Firstly, it is possible that a mental disorder and the brain lateralisation pattern develop independently from each other since they are controlled by separated sets of genes (Figure 2(1.)). This is conceivable since not all aspects of the neuronal organisation display left–right differences. Nevertheless, both traits can be sensitive towards the same epigenetic factors, like for instance, steroid hormones, immune factors, or environmental experiences—all factors that have a broad impact on developmental processes (Table 1) [63,155,158,177]. Correlated impairments, then, simply reflect parallel disturbed neurodevelopmental processes in response to aversive experiences, which do not necessarily touch the same neuronal networks. In a comparable way, impairments of both traits might independently contribute to the development of cognitive deficits.

Secondly, it is possible that the same genes that control a mental disorder also influence the cerebral lateralisation pattern. This means that there are cellular mechanisms that play a role in the development of both traits, and both traits can be susceptible to the same epigenetic factors. However, parallel effects could simply be detectable because both have a common cellular origin but not since they cause each other (Figure 2(2.)). Recent studies identified common genetic factors (Table 1) for the development of cerebral asymmetries and disorders, such as schizophrenia or autism, like PCSK6 (proprotein convertase subtilisin/kexin type 6) [129,132,133,134], FOXP2 (forkhead box P2 gene) [118,119,120], DRD2 (dopamine 2-receptor) [115,116,117], and LRRTM1 (leucine-rich repeat transmembrane neuronal 1) [122,123,124]. For all genes, however, it is not known which, if any, developmental processes could potentially link the developmental pathways of both traits.

Thirdly, it is conceivable that there are genes that primarily influence the lateralisation of the brain but then have secondary effects on the pathogenesis of a mental disorder (Figure 2(3.)). As explained above, this can generally mean that critical genetic and non-genetic factors exert their effects in a lateralised manner due to the asymmetrical neuronal substrate. The other way around, it is also conceivable that lateralised brain processing buffers the action of factors triggering the emergence of a mental disorder. More specifically, however, it could also mean that the symptoms of a disorder only occur when the typical asymmetrical organisation of critical networks is disrupted. As indicated above, this could, in particular, play a role in the development of cognitive impairments. An example of this could be the association between the degree of hemispheric lateralisation and the experience of hallucinations in schizophrenia, as schizophrenic patients with auditory hallucinations display reduced language lateralisation [61].

Fourthly, it is conceivable that certain genes primarily control the development of a mental disorder since they impair neuronal functioning, synaptic communication, or differentiation processes. As a consequence of the resulting disturbed information processing, the normal pattern of hemispheric lateralisation can also be disturbed (Figure 2(4.)). Animal research has shown that the stabilization of brain asymmetries depends on activity-dependent modulations of higher order intra- and interhemispheric neuronal networks [177,179]. An example could be the use of dichotomous information processing strategies. In general, the right hemisphere analyses more fine stimulus details (local processing style), while the right hemisphere is specialized in analysing stimuli in an overall spatial–relational configuration (global processing style) [230,231]. In patients with schizophrenia, for instance, global information processing is compromised [232], presumably due to impairments in the magnocellular visual pathway [233,234,235,236,237]. Interestingly, in particular, normally lateralised cognitive processes, such as the discrimination of faces and other body parts [234,237,238,239,240], as well as spatial memory and orientation performance [241,242,243], which benefit from a global analysis strategy, are impaired. It is, therefore, conceivable, although not yet tested, that sensory impairments prevent the emergence of a hemisphere-specific division of dichotomous processing strategies in schizophrenia.

The complexity of potential intertwined interactions is illustrated by the action of ontogenetic stress experience [100,147]. Stress has a widespread influence on the pathogenesis of different mental disorders during pre- and postnatal development [62,100,156,244,245,246,247]. When pregnant rodent dams are exposed to chronic stress, they release increased amounts of glucocorticoids, which enter the embryonic organism via the placenta and influence the physical and neuronal development of the offspring, including changes in stress reactivity that, in turn, increase the risk of developing a mental disorder [155,248]. In parallel, ontogenetic stress affects cerebral lateralisation patterns [100,151]. For instance, the likelihood of a child being mixed-handed increases with the number of traumatic events experienced by the mother during late pregnancy [249]. First studies in preclinical rodent models have also demonstrated changes in motor asymmetries following early stress experiences [250]. On the other hand, the mature neuronal stress system is organised asymmetrically, whereby the degree of cerebral lateralisation influences emotional and physiological reactivity to stress [147,251,252,253,254], and current stress affects behavioural lateralisation. This implies that lateralised stress processing has a protective effect against inappropriate stress responses [147,255] and, thus, may counteract the development of a mental disorder [147]. These observations show, first of all, that stress can influence the development of lateralisation and the pathogenesis of a mental disorder in parallel. There are a multitude of different neuronal populations that are sensitive to stress-mediating factors, such as glucocorticoids [100] (Figure 2(1.)). If there are ontogenetic connections between both traits, these are probably not unidirectional but complex and mutual. Thereby, the extent and direction of possible inter-relations could depend on the specific processing levels and/or developmental phases.

An additional psychological factor that may mediate the role of stress in the development of brain lateralisation and a mental disorder could be personality and coping style. On the one hand, certain personality traits are associated with a higher risk of mental disorders. For example, high psychoticism and neuroticism (or generally, negative emotionality) is a susceptibility factor for affective and anxiety disorders [98,256,257,258]. A negative emotionality trait is associated with changes in the lateralised structural organisation of the corticolimbic system [259] and its functional connectivity at rest [260]. On the other hand, there is evidence from human and animal research that behavioural lateralisation, as indicated by handedness, is related to coping style in stressful situations. In an anxiety-inducing situation, left-handed people show greater reactivity compared to right-handed people [261]. Marmots that view an approaching person with their left eye are also more likely to flee than individuals that perceive this potential threat with their right eye [262]. Correlations between personality, stress reactivity, and handedness have also been observed in various monkey species [263].

## 4. Potential Research Approaches in Animal Models

In order to clarify the extent to which the development of cerebral asymmetries and mental disorders influence or even cause each other, experimental research into the sequence and direction of possible interactions, as well as the mediating cellular mechanisms, in suitable animal models is required. Due to the widespread use of brain lateralisations, a wide range of animal species are available for this purpose [2]. Some major animal models are already being used in both research fields (Table 2).

The zebrafish *Danio rerio*, for example, is an important model of asymmetry research, in which the genetic and cellular foundations of structural and functional left–right differences are investigated in particular [264,265,266,267]. On the other hand, it has also been established as a model of complex brain disorders and drug-induced conditions [268,269]. Even invertebrates like the nematode *Caenorhabditis elegans* or the fruit fly *Drosophila melanogasta* [270], which show asymmetries in neuronal organisation and behaviour, are currently being used to explore the genetic and cellular foundations of mental disorders like autism [271,272,273,274] or schizophrenia [275,276,277]. Rodents, such as rats and mice in particular, serve as proven translational models for the development of psychopathological disorders, in which the influence of aversive factors such as stress or immune stimulation, among other things, are investigated [146,278,279]. They show asymmetries of the brain and behaviour that are modulated by environmental conditions and sex-dependent factors [165,280,281,282,283]. Rodents display individual paw preferences [284] and show an intrinsic turning bias that can be modulated by the experimental manipulation of striatal dopamine release [285]. The first studies with rodent models for psychopathological diseases show that genetic manipulations [286] or ontogenetic stress experiences influence the expression of motor asymmetries [287,288,289,290]. These effects are presumably mediated by the ontogenetic modulation of dopaminergic signalling. For instance, maternal immune activation (MIA), which is a developmental animal model for schizophrenia, leads to reduced DRD2 mRNA in adolescent offspring, which is accompanied by increased a right-side turning preference [290]. In rats, stress processing is lateralised, with a dominant right-hemispheric activation of prefrontal cortical areas, while the left side can counteract right-sided dominance via interhemispheric inhibitory mechanisms. This pattern of lateralised stress regulation is possibly related to efficient stress and self-regulation and is shaped by ontogenetic experiences and sex-dependent factors [152]. Thus, exploring the underlying cellular and functional mechanisms in rodent models, especially, could serve as a blueprint for understanding the potential associations between stress reactivity, lateralisation patterns, and the development of mental disorders.

**Table 2 brainsci-15-00169-t002:** Major animal models in asymmetry and preclinical research.

Model Species	Asymmetry Research	Preclinical Research
Round worm (*Caenorhabditis elegans*)	Genetics of structural/functional left–right differences in neuronal networks [270,291]. Experience-dependent effects onto lateralised information processing and behaviour [292].	Genetics and functioning of conserved neurochemical pathways associated with mental disorders, drug responses [273,275,276,293].
Fruit fly (*Drosophila melanogasta*)	Genetics of structural/functional left–right differences in neuronal networks. Structure—function inter-relationships between neuronal asymmetries and information processing and behaviour [270].	Genetics/epigenetics and functioning of conserved neurochemical pathways associated with mental disorders, drug responses, behavioural phenotypes [271,274,276,293]. Cellular mechanisms mediating gut–microbiome–brain axis communication relevant for mental disorders [272].
Zebra fish (*Danio rerio*)	Genetic and epigenetic pathways of structural/functional brain asymmetries [177,265,267,294]. Relationship between neuronal (epithalamus) asymmetries and information processing, personality, emotional behaviour [270,295,296].	Genetics/epigenetics and functioning of neurochemical pathways associated with mental disorders, drug responses, behavioural phenotypes [293]. Valproic acid model for autism [187,297,298].
Rodents: mice (*Mus musculus*), rats (*rattus rattus*)	Genetics and structural, cellular foundations of asymmetries in brain and behaviour [279,281,283]. Impact of environmental, sex-dependent factors on neuronal and behavioural asymmetries [164,165,280,299]. Impact of ontogenetic stress on neuronal and behavioural asymmetries [152,250,288,290,300]. Relationship between transmitter asymmetries and behaviour [301,302,303].	Genetics/epigenetics and functioning of neurochemical pathways and transmitter systems associated with mental disorders or drug responses [276,293,304,305]; complex behavioural phenotypes, including the role of dopamine for hallucinations [306]. Impact of hormonal, immune factors on mental disorders. Cellular mechanisms mediating gut–microbiome–brain axis communication relevant for mental disorders [276,293,304,305]. Role of aversive factors (MIA [145,146,278,307,308]; stress [155,156,244,309,310,311]; valproic acid [312,313]).
Birds: chicken (*Gallus gallus*), pigeon (*Columba livia*), quail (*Coturnix japonica*)	Role of hormones (corticosterone, testosterone, estradiol), sensory experience (light) on structural and functional brain asymmetries. Impact of lateralised processing on cognitive performances and interhemispheric communication [34,167,177,179,231].	Chicken as valproic acid model for autism [314,315,316] especially for impaired social cognition [317,318]. Chicken as an anxiety and depression model [319,320,321,322]. Corticosterone level manipulation in hens or eggs of as prenatal stress model [323,324]. Role of the microbiota–gut–brain axis for stress-induced injurious behaviours [325,326].

MIA = maternal immune activation.

Possible interactions between asymmetry formation and the pathogenesis of a mental disorder are certainly complex and differ between specific disorders (Figure 2, Table 1). In order to elucidate the degree and direction of potential causal associations, it is necessary to analyse in which ontogenetic time windows interactions can take place, which genetic and epigenetic factors are involved, and which cellular mechanisms mediate these interactions and, thereby, determine the structural and functional effects. This gives rise to a number of research questions which could be answered using suitable animal models in particular. Some of the most important research approaches areDevelopmental pathways linking asymmetry formation and the pathophysiology of a mental disorder.The temporal sequence of asymmetry formation in relation to early pathophysiology: To understand the direction of possible developmental relationships, we need information about the temporal sequence of asymmetrical differentiation processes, specifically in the brain areas relevant to a specific mental disorder. We have to know whether asymmetries at the cytoarchitectonic, connectivity, and/or neurochemical level develop before or after the first signs of developmental abnormalities that characterise the pathogenesis of a mental disorder (Figure 1).Convergence of developmental pathways: To test whether the development of both traits is causally linked, we need to know in which ontogenetic phases the developmental pathways of cerebral asymmetries and mental disorders actually overlap and how these are influenced by the interaction of distinct ontogenetic factors (Figure 2). Table 1 provides a number of factors that could play such an integrating role. In addition to the expression pattern of critical genes, the synergistic effects of endogenous factors, such as sex hormones or immune factors, on the one hand, and environmental experiences, on the other, could be crucial. Studies on the lateralised visual system of chickens, for example, have shown that the decisive role of asymmetrical light experiences shortly before hatching for the development of asymmetries is modulated by various steroid hormones [34,168]. By investigating the additive effect of distinct factors at critical developmental times points, causal relationships in the sequence of neuronal developmental steps can be uncovered and the underlying neuronal mechanisms explored. Currently, we have only very limited knowledge about potential cellular pathways.Role of ontogenetic plasticity in linking asymmetry formation and the pathogenesis of a mental disorder.Since ontogenetic experiences play an important role in the development of both traits, it likely that neuronal activity-dependent mechanisms mediating ontogenetic plasticity contribute causally to their intertwined development. This idea is supported by the observation that genes like DRD2 [115] or LRRTM1 [327], which play a role for both traits (Table 1), are involved in synaptic plasticity.(a)Neuronal mechanisms of neuronal plasticity: To understand whether the neuronal mechanisms of ontogenetic plasticity play a causal role for linking the development of both traits, it is necessary to explore the potential lateralisation of these mechanisms in response to critical aversive influences and genetic risk factors. A central key player could be a neurotrophic factor like BDNF (brain-derived neurotrophic factor; Table 1), which regulates activity-dependent synaptic stabilization, axo-dendritic growth, arborisation, or cell survival [328]. Research in birds has shown, for example, that a structure-specific asymmetrical activation of the BDNF signalling cascade is associated with the light-dependent development of visual asymmetries [106,329]. On the other hand, BDNF mediates aversive stress effects [113,330], and the single nucleotide polymorphism of the BDNF gene leads to a differential vulnerability to mental disorders [258,331]. It is possible that insufficient neurotrophic support contributes to impaired neuronal development and, hence, to a suboptimal organisation of neuronal networks in the first place and impairs the ability of neuronal networks to react adaptively to aversive experiences [332].(b)Influence of environmental experiences: To test whether impaired processing mechanisms, as they characterise a psychopathological disorder, influence the development of cerebral asymmetries (Figure 2(3.)), it could be useful to explore, specifically, the role of sensory stimulation. Sensory experiences play a crucial role in regulating developmental trajectories and the differentiation of both sensory systems, in particular, and the overarching organisation of the brain [333,334,335], including its lateralisation [167,177,179,231]. Research in birds and zebrafish has shown that the development of structural and functional asymmetries initially begins in the sensory processing pathways but later in development affects higher-order intra- and interhemispheric processing and, in turn, lateralised sensorimotor control and decision making [167,168,177,203,294,336,337]. In congenitally blind humans, left lateralisation in frontotemporal language areas is reduced [338,339]. Thus, when sensory processing and/or plasticity is impaired, as observed, for example, in schizophrenia [232,233,235,237], it is possible that these impairments also affect asymmetry formation and, secondarily, cognition. In this context, research with birds can serve as a blueprint for research into corresponding developmental chains in other preclinical models [179].Relationships between lateralisation patterns and cognitive deficits.As suggested above, higher cognitive deficits accompanying mental disorders may arise only as secondary consequences of disrupted intra- and interhemispheric processing, i.e., from a disorganised lateralised functional architecture of brain networks (Figure 2). To better understand the relationships between lateralised information processing and the cognitive consequences associated with a mental disorder, it would be useful to compare the cognitive performance of preclinical animal models, in which the experimental manipulation of hemispheric asymmetries is possible. Research in birds, for instance, has shown that interhemispheric communication is critically impaired in non-lateralised individuals [34,337,340,341].

## 5. Conclusions

Overall, there is ample evidence that the ontogenetic pathways underlying the development of a mental disorder and an atypical lateralisation pattern may be intertwined. This implies that both traits are more than just two parallel-developing characteristics of the functional architecture of the brain [39,342]. However, we need much more research using appropriate basic and translational models to understand the direction, extent, and timing of potential interactions from the cellular to the functional level. Elucidating these relationships could then provide fundamental insights into the aversive and protective developmental processes underlying normal and psychopathological development.

Altered cerebral asymmetries could causally contribute to the development of symptoms of a mental disorder, as neural mechanisms that trigger the pathogenesis of a disorder are embedded in the asymmetrical organisation of the developing brain. Therefore, the occurrence and severity of impairments in neural processing and cognition can probably not be understood independently of the development of the lateralised organisation of intra- and interhemispheric neural networks. In this sense, we have to explore whether mental pathogenesis can be interpreted as two brain hemispheres developing out of balance.

## Figures and Tables

**Figure 1 brainsci-15-00169-f001:**
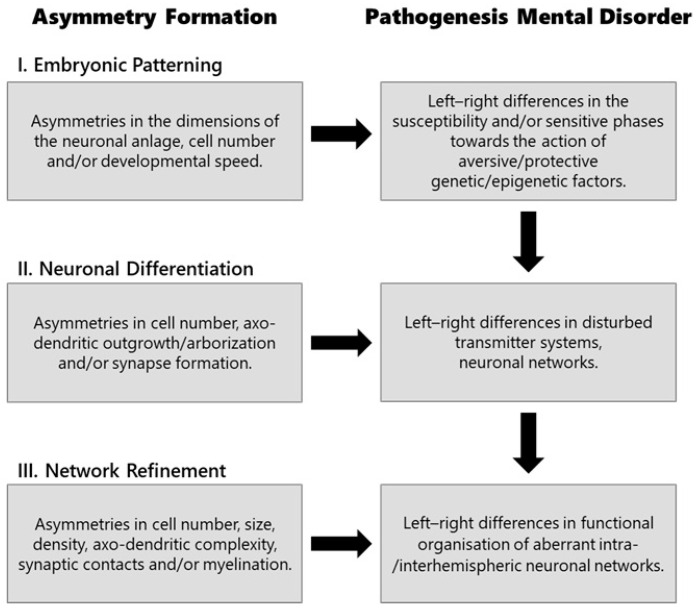
Potential effects of asymmetry formation on the pathogenesis of a mental disorder during the three main phases of neuronal development—each phase is dominated by different cellular processes that lead to an increasingly lateralised functional organisation of the two cerebral hemispheres [179]. As a consequence, factors and processes that regulate the pathogenesis of a disorder act in a lateralised manner.

**Figure 2 brainsci-15-00169-f002:**
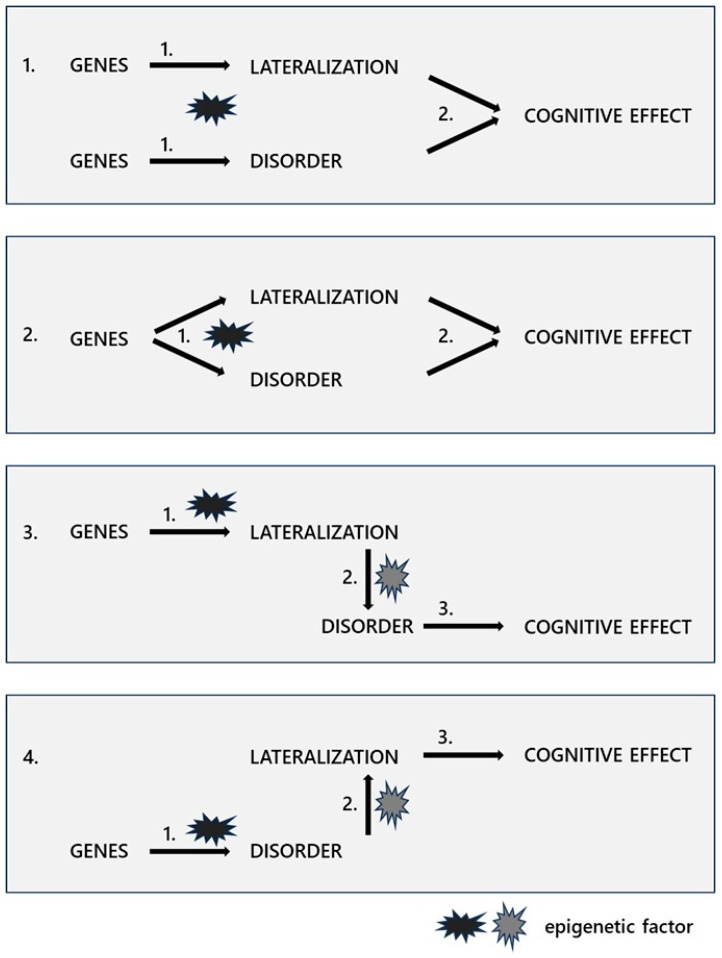
Simplified scheme of four possible links between the development of hemispheric asymmetries (LATERALISATION) and the pathogenesis of a mental disorder (DISORDER), based on a model proposed by Bishop [211]. While genetic factors (GENES) determine the susceptibility to the development of a particular mental disorder on the one hand and to hemispheric asymmetries on the other, it is epigenetic factors (stars), which have a decisive influence on development during different ontogenetic phases and may play a linking role between developmental pathways. The different modes of genetic–epigenetic interactions determine the properties of synaptic transmission and plasticity, as well as the structural and functional organisation of neuronal networks, leading to adaptive or maladaptive functionality and COGNITIVE EFFECTS. Not all functional consequences are related between both traits; here is only indicated how a cognitive effect results from the developmental pathways described. The direction and extent of the relationships can vary depending on the neuronal subsystems or cognitive modules and developmental phases involved.

**Table 1 brainsci-15-00169-t001:** Examples of genetic and non-genetic factors identified to affect the development of both brain asymmetries and mental disorders.

**Genetic Factors**		
Arrestin beta 2 (*ARRB2*). Activator of transcription and developmental regulator (*AUTS2*). BAR/IMD domain containing adaptor protein (*2BAIAP2*). Frizzled class receptor 9 (*FEZ1*). Growth-associated protein 43 (*GAP43*). Roundabout 1 (*ROBO1*).	Transient asymmetrical expression in zebrafish [103] and in mouse/human forebrain (*2BAIAP2*, *GAP43*) [104].	Arrestin associated with ASD or ADHD in humans [103].
Brain-derived neurotrophic factor (*BDNF*).	Asymmetrical expression in developing rodent hippocampus [105]. Asymmetrical activation of BDNF signalling cascade in response to sensory stimulation [106].	Polymorphism associated with affective disorders [107,108]. Polymorphism associated with cognitive impairments in affective disorders [109,110,111,112]. Downstream mediator for environmental factors, like stress [113,114].
Dopamine 2-receptor gene (*DRD2*).	Asymmetrical expression within different cortical areas in humans [115].	Polymorphism associated with SZ [116,117].
Forkhead box P2 gene (*FOXP2*).	Polymorphism associated with interindividual variability in hemispheric asymmetries for speech perception [118].	Polymorphism associated with SZ [119,120]. Associated with cognitive impairments in SZ [121].
Leucine-rich repeat transmembrane neuronal 1 (*LRRTM1*).	Polymorphism associated with handedness [122,123,124].	Polymorphism associated with ASD and SZ [124,125].
Lim domain only 4 (*LMO4*).	Transient asymmetrical expression in embryonic (perisylvian) human/mouse cortex [104]. Modulation of lateralised expression modifies paw preference in mice [126]. Involved in patterning thalamocortical connections in zebrafish [103].	Polymorphism in promoter region associated with SZ [127]. SZ-like behaviours in Lmo4-deficient mice [128]. ASD-like behaviours in Lmo4-deficient mice [129,130].
Proprotein convertase subtilisin/kexin type 6 (*PCSK6*).	Involved in left–right patterning of body and brain [131]. Polymorphism associated with handedness [129,132,133] and structural asymmetries in temporal cortical areas [134].	Polymorphism associated with ASD and schizotypy in sex-dependent manner [133].
**Epigenetic Factors**		
Immune factors.	Interindividual differences in immune reactivity related to paw preference and stress reactivity in mice [135,136,137]. Association between lateralisation of frontal brain activity, variation in affective style, and immune function in adult humans [138,139,140], especially after childhood maltreatment [138].	Mental disorders associated with immune dysregulation [141,142]. Maternal infection increases risk for psychiatric disorder in the offspring. Rodent MIA models indicate activation of inflammatory pathways resulting in increased levels of cytokines and chemokines. MIA-induced microglia alterations linked to development of schizophrenia- and autism-like behaviours [143,144,145,146]. Psychosocial stress as a critical modulator of neuroimmune activity [143].
Pre/postnatal stress exposure.	Changes in structural and functional (e.g., handedness) brain asymmetries, personality, and emotional reactivity in humans and other mammals [147,148,149,150,151,152].	Risk factor for mental disorders and associated cognitive impairments [145,150,153,154]. Via stress-induced maternal increase of glucocorticoid affecting foetal brain development [100,155], altered epigenetic regulation [156,157]. Alterations in the HPA axis associated with modification of transmitter systems (dopamine, serotonin) in mental disorders [100].
Month of birth (climate environment).	Affects hand preference [158] (higher prevalence of left-handedness for birth during the spring/early summer; linked to factors like maternal infection, length of photostimulation, nutrition) [159]).	Affects risk against affective disorders (linked to factors like maternal infection, nutrition (vitamin D), length of photostimulation) [144,160,161,162]. Increased risk for schizophrenia born in winter/spring in northern hemisphere [163].
Sex-related factors.	Sex differences in neuroanatomical left–right differences in humans and mice [164,165]. Higher prevalence of left-handedness in males [123,166]. Sex hormones affect lateralised brain physiology, intra/interhemispheric communication, and cognitive functions [2,94]. Prenatal and pubertal testosterone level influences strength of brain asymmetries in humans and rodents [2]. Sex hormones modify action of sensory stimulation in chickens [34,167,168].	Sex/gender differences concerning risk, prevalence, age of onset, and symptomatology in several mental disorders [169,170,171]. A low level of oestradiol (progesterone) increases the risk and severity of affective disorders in females [172]. High foetal testosterone level increases risk of ASD, correlates with autistic features [173], potential protective benefits against affective disorders [174]. Sex-specific differences in neuroendocrine and immunological activity [175] due to interaction of sex-specific risk genes with sex-sensitive signalling pathways [176].
Sensory experience.	Biased sensory input (caused by perinatal asymmetries) influences structural and functional brain asymmetries [177,178,179]. Ontogenetic light stimulation affects development of visual asymmetries in birds [34,167,177,179].	Potential effects of perinatal light exposure onto risk of depression [162].
Socio-cultural effects.	Cultural pressure for right-handedness [158,180,181]. Social encouragement affects hand preference in human infants [182].	Socioeconomical effects on aetiology of mental disorders [183,184,185,186].
Teratogens: valproic acid (*VPA*).	Modifies lateralisation of social behaviour, size of epithalamic nuclei, asymmetrical expression of epithalamic marker genes in zebra fish [187].	Increases risk for neurodevelopmental disorders like ASD and ADHD by affecting DNA methylation [188].
Retinoic acid (*RA*).	Involved in early left–right patterning of body and brain [189]. Involved in left–right differentiation of neuronal cell populations [190].	Affects cognitive dysfunction in SZ and pathophysiology of ASD by affecting molecular patterning and connectivity pattern of forebrain areas [191].

(ASD = autism spectrum disorder; ADHD = attention deficit hyperactivity disorder; HPA = hypothalamic–pituitary–adrenal axis; MIA = maternal immune activation; SZ = schizophrenia.)

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
