# Peer review of "The Balance in the Head: How Developmental Factors Explain Relationships Between Brain Asymmetries and Mental Diseases"

_brainsci, 2025, doi:10.3390/brainsci15020169_

Round 1
Reviewer 1 Report
Comments and Suggestions for Authors
The present work focuses on the relationship between lateralization in the brain and psychiatric disorders. It aims to inform on the topic, but the research question to be addressed is too general and wide to be readily adopted by readers. First, what kind of lateralization is meant by the authors should better be defined. Second, the evidence on the topic is not definite and the authors should better acknowledge the uncertainty on reported associations / causal links. Following, a list of comments that may be addressed in order to improve the manuscript:
- Abstract:
Lines 18-19. Most psychiatric disorders are not associated with cognitive impairments, or cognitive impairments precede mental disorders / influence clinical presentations solely as risk factors and not as part of the diagnostic entity.
"Altered cerebral asymmetries may contribute causally to the development of symptoms of a psychiatric disorder, as neural mechanisms that trigger the pathogenesis of a disorder are embedded in the asymmetric organisation of the developing brain."
Lines 19-20 "Recent research has identified genetic and epigenetic factors that influence both the pathogenesis of mental illness and the development of brain asymmetry patterns" These identified factors are not described in the main text. Please better address these factors, and report how they were identified. A table in the main text with references would greatly improve the reliability of this statement, and would allow readers to translate the contents of the review to their specific field of interest. A systematic approach to the retrieval of such factors would also increase the significance of the report.
Lines 25-27. How are these mechanisms causally linked to disorders? What specific neural mechanisms? How definite and causal has been proven to be the relationship between brain asymmetry and psychiatric disorders/symptoms? To my understanding, these associations are weak to moderate at best, and only limited evidence supports this relationship.
Line 31-32 "In this sense, mental pathophysiology could be interpreted as the result of two hemispheres of the brain that are out of balance." This sentence is not supported by sufficient evidence. I would not be fully comfortable reporting such a conclusion as summarizing the scientific consensus on the topic. Please revise.
- Introduction:
The introduction should further address
1) the age of onset of mental disorders (https://doi.org/10.1038/s41380-021-01161-7)
2) how most cellular processes described as altered in mental disorders (not all mental disorders, but mostly schizophrenia and autism, by defective or excessive pruning, as well as altered neuronal migration) occur way before the age of onset. These processes do not substantially modify during adolescence, although a second phase of development may be described (https://doi.org/10.1002/dev.22366). How do the authors postulate that these two traits then interact? This is later discussed, but a reader would be interested in a first interpretation.
Moreover, the difference between functional or structural asymmetry should better be described, and the degree of uncertainty whether observed associations are either causal and/or substantially influencing clinical presentations better acknowledged. It would be worthy of mentioning that stroke lesions in language areas are now understood as more complex than first described by Broca, and a high heterogeneity can be observed between individuals (not all exhibit a left-lateralization, even among right-handed individuals). Therefore lines 75-76 should be revised, as the concept of a "prototypical division" between left and right hemispheres is not fully backed by scientific evidence. The authors indeed state so later "left and right hemispheres are dominant for the same functions in a majority (...)" line 76-77
It would also be important to mention that most significant asymmetries in psychiatric disorders have arguably been observed in neurodevelopmental disorders, such as autism spectrum disorders (https://doi.org/10.1038/s41467-019-13005-8) or ADHD (https://doi.org/10.1007/s11682-022-00708-8).
- Section 1
It would also be advised to better revise molecular mechanisms underlying brain maturation, neuronal migration, and which pathways control for the development of physiological lateralization or have been implicated in influencing altered lateralization in psychiatric disorders.
The association between handedness and lateralization has also been the interest of recent research, which, however reports mixed findings between these traits - especially structural asymmetry (https://doi.org/10.1038/s41467-024-46690-1; https://doi.org/10.1016/j.neubiorev.2014.04.008). Please revise lines 93-100 accordingly.
Figure 1 - What do the authors mean by "patterning"?
Molecular mechanisms underlying I. / II. and III. reported in Figure 1 should be described in more detail in the text, which at the present time appears too generic to inform a biochemical approach to the topic (e.g. "genes" is too wide of a category - which pathways? Regulated by which factors? Related to which symptoms if altered?).
Figure 2, a fifth way has not been considered. That genes inform the risk for a disorder, and the disorder itself is associated with a divergent functional lateralization - which may either be adaptive or maladaptive in nature.
- Section 3
A third figure summarizing animal models would be highly appreciated by readers - and may enrich the section which is interestingly and impactful.
The five research questions may better be highlighted - either with an additional figure or a more readable list (to consider if a non-numbered list would be more beneficial, with a first summary of the five research questions and more details following later).
- Section 4
In conclusions, the authors seem to inflate psychiatric diagnoses with psychopathological disorders. The two are not synonymous. The potential for lateralization features to be adopted as biomarkers is in its infancy at best - at the present moment they are neither diagnostic nor inform clinical treatments or prognosis.
- Other comments:
Authors should revise the manuscript for consistent language and editing throughout sections. For instance lines 36-55 have one-space interline, and subsequent lines have text double space. Later in the text single space is adopted again.
Author Response
Responses to reviewer 1:
The present work focuses on the relationship between lateralization in the brain and psychiatric disorders. It aims to inform on the topic, but the research question to be addressed is too general and wide to be readily adopted by readers. First, what kind of lateralization is meant by the authors should better be defined. Second, the evidence on the topic is not definite and the authors should better acknowledge the uncertainty on reported associations / causal links. Following, a list of comments that may be addressed in order to improve the manuscript:
We thank the reviewer for pointing out aspects that were not clearly presented in our original manuscript. Taking into account the suggestions below, we have carefully revised the manuscript and clarified the issues raised by adding details and explanations as well as two new tables. In this context, we have included a considerable number of additional references.
- Abstract:
Lines 18-19. Most psychiatric disorders are not associated with cognitive impairments, or cognitive impairments precede mental disorders / influence clinical presentations solely as risk factors and not as part of the diagnostic entity./ "Altered cerebral asymmetries may contribute causally to the development of symptoms of a psychiatric disorder, as neural mechanisms that trigger the pathogenesis of a disorder are embedded in the asymmetric organisation of the developing brain." Lines 25-27. How are these mechanisms causally linked to disorders? What specific neural mechanisms? How definite and causal has been proven to be the relationship between brain asymmetry and psychiatric disorders/symptoms? To my understanding, these associations are weak to moderate at best, and only limited evidence supports this relationship./ Line 31-32 "In this sense, mental pathophysiology could be interpreted as the result of two hemispheres of the brain that are out of balance." This sentence is not supported by sufficient evidence. I would not be fully comfortable reporting such a conclusion as summarizing the scientific consensus on the topic. Please revise.
To avoid unclarities, we have rewritten the abstract with more careful wording and omitted the last statement since it might be too speculative in short words.
Lines 19-20 "Recent research has identified genetic and epigenetic factors that influence both the pathogenesis of mental illness and the development of brain asymmetry patterns" These identified factors are not described in the main text. Please better address these factors, and report how they were identified. A table in the main text with references would greatly improve the reliability of this statement, and would allow readers to translate the contents of the review to their specific field of interest. A systematic approach to the retrieval of such factors would also increase the significance of the report.
Thank you for this suggestion. We now include the new table 1 summarizing the impact of specific genetic- and non-genetic factors onto asymmetry formation and pathophysiology of mental disorders. This table illustrates the variety of different factors which affect both traits and how they could potentially interact.
- Introduction:
The introduction should further address
1) the age of onset of mental disorders (https://doi.org/10.1038/s41380-021-01161-7)
2) how most cellular processes described as altered in mental disorders (not all mental disorders, but mostly schizophrenia and autism, by defective or excessive pruning, as well as altered neuronal migration) occur way before the age of onset. These processes do not substantially modify during adolescence, although a second phase of development may be described (https://doi.org/10.1002/dev.22366). How do the authors postulate that these two traits then interact? This is later discussed, but a reader would be interested in a first interpretation.
It would also be important to mention that most significant asymmetries in psychiatric disorders have arguably been observed in neurodevelopmental disorders, such as autism spectrum disorders (https://doi.org/10.1038/s41467-019-13005-8) or ADHD (https://doi.org/10.1007/s11682-022-00708-8).
Thank you for indicating these unclear aspects. We have added a section addressing this question. Here we also include the observation that especially developmental disorders are accompanied by modified brain asymmetries (lines 187-205).
Moreover, the difference between functional or structural asymmetry should better be described, and the degree of uncertainty whether observed associations are either causal and/or substantially influencing clinical presentations better acknowledged.
Thank you, this is a relevant aspect and we provide some more details concerning these uncertainties (lines 98-100, 137-139, 165-186).
It would be worthy of mentioning that stroke lesions in language areas are now understood as more complex than first described by Broca, and a high heterogeneity can be observed between individuals (not all exhibit a left-lateralization, even among right-handed individuals). Therefore lines 75-76 should be revised, as the concept of a "prototypical division" between left and right hemispheres is not fully backed by scientific evidence. The authors indeed state so later "left and right hemispheres are dominant for the same functions in a majority (...)" line 76-77
Thank you for this comment. We have revised the relevant sentences added references that indicate the variability in lateralisation patterns (lines 67-70, 79-87).
- Section 1
It would also be advised to better revise molecular mechanisms underlying brain maturation, neuronal migration, and which pathways control for the development of physiological lateralization or have been implicated in influencing altered lateralization in psychiatric disorders.
This is a good suggestion. We therefore indicated potential pathways at different appropriate positions in the revised text and in the new table 1.
The association between handedness and lateralization has also been the interest of recent research, which, however reports mixed findings between these traits - especially structural asymmetry (https://doi.org/10.1038/s41467-024-46690-1; https://doi.org/10.1016/j.neubiorev.2014.04.008). Please revise lines 93-100 accordingly.
Thank you for pointing to this point. We included this observation into the new section discussing unclarities in structure-functions interrelations (lines 170-173).
Figure 1 - What do the authors mean by "patterning"?
We apologize for this vagueness and explain the term in more detail in the main text (lines 280-282, 286-288).
Molecular mechanisms underlying I. / II. and III. reported in Figure 1 should be described in more detail in the text, which at the present time appears too generic to inform a biochemical approach to the topic (e.g. "genes" is too wide of a category - which pathways? Regulated by which factors? Related to which symptoms if altered?).
We understand that the reviewer asked for more details about genetic and cellular pathways. However, it was not the aim of this report to provide an overview about the cellular mechanisms, we want to concentrate on principal associations. However, we included more details concerning potential neuronal pathways when describing the three developmental steps (lines 297-311). Table 1 includes some additional information.
Figure 2, a fifth way has not been considered. That genes inform the risk for a disorder, and the disorder itself is associated with a divergent functional lateralization - which may either be adaptive or maladaptive in nature.
Thank you for this idea. However, we are not sure whether this way is not the one included in model 4. We now have modified the models including explicitly functional effects and explain interrelations in more detail (figure 2 and legend lines 329-334).
A third figure summarizing animal models would be highly appreciated by readers - and may enrich the section which is interestingly and impactful.
Thank you for this good suggestion. We therefore include the new table 2 summarizing how major animal model are used in asymmetry and preclinical research. In our opinion, this overview clearly illustrates the potential of integrated research approaches.
The five research questions may better be highlighted - either with an additional figure or a more readable list (to consider if a non-numbered list would be more beneficial, with a first summary of the five research questions and more details following later).
We agree with the reviewer that the research questions were not explained clearly enough, so we have completely rewritten the section (integrating information from previous sections) to better explain the rationale for the research questions proposed here (lines 464-543).
- Section 4
In conclusions, the authors seem to inflate psychiatric diagnoses with psychopathological disorders. The two are not synonymous. The potential for lateralization features to be adopted as biomarkers is in its infancy at best - at the present moment they are neither diagnostic nor inform clinical treatments or prognosis.
The reviewer is certainly correct that lateralisation as a biomarker for mental disorders is currently no more than a hypothesis, and we did not mean to imply that. We have therefore amended the relevant wording (lines 548-549; lines 559-561).
- Other comments:
“Authors should revise the manuscript for consistent language and editing throughout sections. For instance lines 36-55 have one-space interline, and subsequent lines have text double space. Later in the text single space is adopted again.“
Thank you, we carefully edited the manuscript after revising the manuscript.
Reviewer 2 Report
Comments and Suggestions for Authors
The manuscript provides a detailed review of studies on the role of developmental factors in the relationship between hemispheric brain asymmetry with mental disorders. From my point of view, the manuscript is very well prepared. The review was performed at a high methodological level. The review's topic has significant implications for understanding the association between genetic (and epigenetic) factors with brain endophenotype and mental disorders.
I have a minor recommendation to improve the manuscript. It is well known that inclination to mental disorders such as depression and anxiety disorder is positively or negatively associated with psychological personality traits such as neuroticism or extraversion. At the same time, inter-individual differences in these psychological traits in non-clinical subjects correlate with differences in their hemispheric asymmetry. I propose to add to the manuscript a short review on the relationship between hemispheric asymmetry, developmental factors and individual values of the psychological traits, affecting the predisposition to mental disorders (such as extraversion-introversion, neuroticism, etc).
Line 113: “Individuals with ADS may exhibit highly individualised patterns of” Please, check - ADS or ASD? Also “individualized” or “individualized”?
Author Response
see also attachment
The manuscript provides a detailed review of studies on the role of developmental factors in the relationship between hemispheric brain asymmetry with mental disorders. From my point of view, the manuscript is very well prepared. The review was performed at a high methodological level. The review's topic has significant implications for understanding the association between genetic (and epigenetic) factors with brain endophenotype and mental disorders.
Thank you for the very positive evaluation of our work.
I have a minor recommendation to improve the manuscript. It is well known that inclination to mental disorders such as depression and anxiety disorder is positively or negatively associated with psychological personality traits such as neuroticism or extraversion. At the same time, inter-individual differences in these psychological traits in non-clinical subjects correlate with differences in their hemispheric asymmetry. I propose to add to the manuscript a short review on the relationship between hemispheric asymmetry, developmental factors and individual values of the psychological traits, affecting the predisposition to mental disorders (such as extraversion-introversion, neuroticism, etc).
Thank you for pointing out this interesting aspect. Personality could be a mediating factor in the effects of ontogenetic stress. We have therefore added relevant information to the section on the effects of stress (lines 413-426).
Line 113: “Individuals with ADS may exhibit highly individualised patterns of” Please, check - ADS or ASD? Also “individualized” or “individualized”?
We checked spelling and abbreviations throughout the text.